# Reconstruction of the Extensor Apparatus After Total Patellectomy in Orthopedic Oncology: A Systematic Literature Review

**DOI:** 10.3390/jcm14144818

**Published:** 2025-07-08

**Authors:** Edoardo Ipponi, Fabrizia Gentili, Fabio Cosseddu, Antonio D’Arienzo, Paolo Domenico Parchi, Lorenzo Andreani

**Affiliations:** Department of Orthopedics and Trauma Surgery, University of Pisa, Via Paradisa 2, 56124 Pisa, Italy

**Keywords:** patella, quadriceps tendon, patellar ligament, functionality, range of motion, complications, local recurrence

## Abstract

**Background**: Patellar resection is recommended in cases of massive cortical bone disruption or malignancies. Modern literature lacks a consensus surgical reconstruction after total patellectomy. Our study reviews the surgical techniques described in the literature and summarizes the reported functional outcomes and complication rates. **Materials**: We systematically reviewed the existing literature, searching the PubMed, Embase, and Scopus databases for articles published between 1950 and 2024. We recorded age, diagnosis, tumor size, Lodwick classification, soft tissue involvement, and pre-operative fractures for each case or case series. We also recorded the reconstructive approaches. Complications, local recurrences, MSTS scores, and knee range of motion (ROM) were considered when reported. **Results**: Twenty-eight articles met our inclusion criteria. Among these, 4 were case series and 24 were case reports. A total of 47 cases treated with total patellectomy were reviewed. Reconstruction was performed with direct suture in 8 cases, while 17 had local augments, including allograft (10 cases), muscle flaps or transportations (4), autologous bone (1), or a composite (2). Reconstruction was not mentioned in 22 cases. ROM was reported for 17 cases, and the MSTS score was reported for 9 cases. **Conclusions**: In cases of relatively small tissue defects, a direct suture of the extensor apparatus can allow adequate functional recovery. In cases of larger gaps, surgeons should use muscle flaps, transfers, or soft tissue augments. Massive bone and tendon allografts should mainly be considered in cases where the neoplasm was not confined to the patella but extensively involved the patellar ligament or the quadriceps tendon.

## 1. Introduction

Benign and malignant patellar neoplasms are extremely rare accounting for approximately 0.12% of all primary bone tumors [1]. Benign tumors represent up to 85% of all lesions, with giant cell tumors of the bone (GCTB) and chondroblastoma among the most frequent histologic types [1,2,3]. Malignant lesions are far less common but should be considered, especially in older patients [3]. The most common primary malignancies include osteosarcoma and chondrosarcoma [4,5,6]. Cancers with osseous tropism, such as lung and renal carcinomas, can be responsible for secondary localizations within the patella [7].

Patellar tumors may be clinically silent and incidentally diagnosed or present with various combinations of anterior knee pain, palpable swelling, and joint stiffness [1]. Imaging evidence, including X-rays, computed tomography (CT), and magnetic resonance imaging (MRI), is necessary to assess lesions’ characteristics, sizes, and orient diagnosis [3]. Although imaging plays a pivotal role in the diagnostic pathway, histology is needed to establish the final diagnosis. Needle or incisional biopsies, along with the subsequent pathological evaluation ,should represent the peak of the diagnostic process [8].

The histological diagnosis is also fundamental in guiding the treatment of each single lesion. Despite an univocal gold standard, most studies suggest that curettage should be considered the treatment of choice for benign bone lesions confined to the patella [9,10]. However, an intralesional curettage is not adequate in the case of extended lesions with massive bone loss or high-grade malignancies. In such conditions, surgeons are generally called to perform a partial or complete patellectomy to achieve wide resection margins [11]. The sacrifice of the patella and the resulting disruption of the extensor apparatus may impair its functionality, negatively impacting patients’ quality of life [12]. Surgical strategies to replace the patella or restore the anatomical continuity of the extensor apparatus could be necessary to increase the postoperative performance of treated knees. Different authors have proposed several reconstructive approaches in recent decades. However, due to the rarity of patellar malignant and locally aggressive tumors, reports have been limited to case reports and very small case series with few or no data regarding the optimal reconstructive management after patellectomy in orthopedic oncology. Our review aims to provide an organic summary of the surgical reconstructions of the patella and extensor apparatus already reported in the literature, evaluating their effectiveness with a particular focus on their functional outcomes and complication rates.

## 2. Materials and Methods

A systematic review of the literature was performed according to the Preferred Reporting Items for Systematic Reviews and Meta-Analyses (PRISMA) guidelines, using a PRISMA checklist and algorithm. The search algorithm, performed by the PRISMA guidelines, is reported in Figure 1. The systematic review has not been registered (Appendix A). A comprehensive search of the PubMed, MEDLINE, EMBASE, Scopus, and Google Scholar databases using various combinations of the keywords “patella,” “patellectomy,” “replacement,” “tumor,” and “sarcoma.” All the original articles reported on the surgical treatment of patellar bone tumors that required complete patellectomy and the eventual subsequent reconstruction. We included papers published between 1950 and 2024.

Three independent reviewers (E.I., F.G., A.D.A.) conducted the research separately. Only articles from peer-reviewed journals were included. The investigators separately reviewed each publication’s abstract and then closely read all articles, extracting data to minimize selection bias and errors. Inclusion criteria were a diagnosis of bone tumor involving the patella, a total patellectomy, detailed preoperative presentation, and details on the outcome of received treatment. Pre-clinical studies, literature reviews, articles that did not mention nor provide data on surgical treatment, and papers written in languages other than English were excluded. 

All articles were initially screened for relevance by title and abstract, excluding articles without an abstract and obtaining the full-text article if the abstract did not allow the investigators to assess the presence of inclusion and exclusion criteria.

Considering the limited number of articles and the low level of evidence in the few available articles, we included in our study articles ranging from Level I to Level V, as well as detailed case reports. Only cases that received complete patellectomy due to a primary or metastatic bone tumor of the patella were included (Figure 2).

The mean age of the patients and the size of the lesions (larger diameter) were recorded. The radiographic behavior of the neoplasms was classified according to the Lodwick classification when obtainable. The eventual involvement of other soft tissues within the extensor apparatus or the occurrence of pre-operative fractures were taken into account. The type of reconstructive approach performed, when reported, was also assessed. The follow-up of all included cases was considered, and the duration, the eventual occurrence of local recurrences, and the incidence and type of complications were evaluated. We evaluated the postoperative function of the treated extensor apparatus, including the Medical Research Council (MRC) Scale for Muscle Strength and the range of motion when available, and the overall performances of the treated lower limbs, including scoring systems such as the ones by MSTS (Musculoskeletal Tumor Society) and ISOLS (International Society Of Limb Salvage). 

Twenty-eight articles were included, comprising four case series and 24 case reports published between 1950 and 2024. 

## 3. Results

Twenty-eight articles met our inclusion criteria [13,14,15,16,17,18,19,20,21,22,23,24,25,26,27,28,29,30,31,32,33,34,35,36,37,38,39,40]. Among these, 4 were case series [13,14,15,16], and 24 were case reports [17,18,19,20,21,22,23,24,25,26,27,28,29,30,31,32,33,34,35,36,37,38,39,40]. A total of 47 cases were included in our analysis. Patients’ mean age was 48.0 (13–82). Thirty-seven cases suffered from primary bone lesions. Sixteen cases were diagnosed with giant cell tumors of the bone, and one of them had a secondary aneurysmal bone cyst (ABC). Three cases had primary ABC, and as many had chondroblastomas. Two other cases were treated for benign lesions like fibrous dysplasia and myopericytoma. Thirteen cases had primary high-grade malignant tumors: 3 osteosarcomas, three synovial sarcomas, three myxofibrosarcomas, two pleomorphic sarcomas, one fibroblastic sarcoma, and one malignant fibrous histiocytoma. Seven cases suffered from the metastatic lesions of the patella. Three of them were lung carcinomas, two had melanoma, one had gastric cancer, and one had breast cancer. The remaining three cases had lymphoma. Thirty-four lesions could be classified according to the Lodwick classification. Among these, 5 cases were of type I (1 IA, 2 IB, 2 IC), 14 were of type II, and 15 were of type III. The larger diameter of neoplastic lesions, reported for nine cases, was, on average, 6.6 cm (2–13). The neoplastic lesions extended to the soft tissues of the extensor apparatus in 10 cases (Figure 3). Five cases had pre-operative fractures. 

The reconstructive approach of choice was described in 25 cases, whereas no details about the reconstructive approach were provided for the 22 remaining cases. What remained of the extensor apparatus after patellectomy was directly sutured in seven cases. In one more case, Gomes et al. [29] sutured the quadriceps tendon with the patellar ligament and reinforced the apparatus with semitendinosus and gracilis tendon grafts. Three cases were treated with muscular flaps. Connell et al. [20] used a myocutaneous flap. Cetinkaya et al. [28] reported on the vastus medialis obliquus advancement, while Osanai et al. [24] associated a gastrocnemius flap with the anterior transposition of the goosefoot tendons. Beyond local flaps and transpositions, autologous iliac crest bone and fascia lata grafts were used by Furuta et al. [35]. Some papers also reported on the use of fascia lata allografts. Chhajed et al. [37] fulfilled the patellar gap and reinforced the extensor apparatus with the iliotibial band and fascia lata allograft. Similarly, Andreani et al. [34] folded a fascia lata allograft to cover a tubulized polyethylene surgical mesh, obtaining a hybrid augment (Figure 4).

Yokoyama et al. [36] proposed a full-synthetic hybrid solution wrapping a synthetic tendon graft around a cement block, mimicking the native patella. The replacement of the entire extensor apparatus, including the patellar bone and tendons, with allografts from cadaver donors was performed in four studies for nine cases (Figure 5).

In five cases, allografts were covered with muscular or musculocutaneous flaps [15], whereas in four cases, bone and tendon allografts were used alone, without skin closure or graft coverage issues. Finally, in 2010, Muramatsu et al. [27] immersed the extensor apparatus in liquid nitrogen, aiming to achieve tumor devitalization, and later reimplanted it in its original position. The complex was then covered by a free vascularized latissimus dorsi myocutaneous flap.

The mean follow-up (the exact duration of which was reported in 39 articles) was 28.2 months (0.4–96). 

The local oncological outcomes of 45 cases were reported in detail. Three of them (6.7%) had local recurrences. 

No infection was reported during this follow-up. Only one case of mechanical rupture was found, as a patient treated with massive bone–tendon allograft reconstruction suffered from a post-traumatic rupture of the quadriceps tendon 6 years after surgery. Only one muscular flap showed partial necrosis [15]. 

The functional outcomes after patellectomy were reported in 10 papers. Four of them reported a post-operative MRC score. Sakuda et al. [40], who used direct sutures after patellectomy without local augmentations, had a MRC score of three out of five. Muramatsu et al. [27] had an MRC score of four after their extensor apparatus cryotherapy underwent external cryotherapy and was sutured back into place. Both Andreani et al. [34] and Cosseddu et al. [38] achieved the complete restoration of extensor apparatus strength (MRC 5) by replacing the patella with a composite consisting of surgical mesh and fascia lata allograft and with a massive bone and ligament allograft, respectively. 

The range of motion of the treated knees was reported in 21 articles. Five cases were reported to have a full post-operative range of motion, but no data were provided regarding the specific motion degrees [14,23,26,32,37]. Similarly, one case was reported to have limited articularity [14] without further data regarding the range of motion. The detailed range of motion was reported in 15 cases [24,25,27,28,29,30,33,34,38,40]. On average, patients had an extension lag of 6.7° (0–10), as complete extension could not be restored in five of them. The mean active flexion was 110 (25–140).

A summary of all included cases is reported in Table 1.

## 4. Discussion

The extensor apparatus is responsible for the knee extension and stabilization of the patellofemoral joint. The patella, in particular, enhances quadriceps action, shifting traction force forward and increasing strength [41]. Patellar tumors should be approached carefully to control or eradicate the disease while preserving the bone’s key role in knee functionality [42]. Intralesional curettage can successfully treat benign and locally aggressive bone lesions confined to the bone itself. However, pathological fractures and massive bone disruptions involving the entire patella could compromise the cortical bone strength and impede a successful curettage [43,44]. In some other cases, even benign tumors can cause the severe deformity of patellar facets, thereby impairing the femoropatellar joint. If the patella is massively damaged and its function and continuity cannot be restored, patellectomy represents a last therapeutic resort [13,14,15,16,17,18,19,20,21,22,23,24,25,26,27,28,29,30,31,32,33,34,35,36,37,38,39,40]. 

Patellectomy should also be considered the treatment of choice for malignant bone tumors that arise from the patella [13,15,16,17,18,19,20,21,24,25,26,27,30,33,35,36,38]. Surgeons are called to perform a resection with wide resection margins to eradicate the neoplasm and allow local disease control. Patellectomy alone can be sufficient for lesions confined to the patella, without significant cortical bone interruptions or pathologic fractures that would spread the tumor to the nearby soft tissues. Malignancies that extend to the quadriceps tendon or the patellar ligament require the sacrifice of the patella and the involved soft tissues. The resection of the extensor chain as a whole can also be necessary in the case of massive tumor spread within the apparatus. If the tumor spreads posteriorly and breaks through the articular facets, dipping into the articular cavity, the whole knee can be contaminated. In this event, patellectomy should be associated with an articular resection and arthroplasty for prosthetic knee reconstruction [15,42].

In our review, the local recurrence rate after patellectomy was 6.7% [13,14,15,16,17,18,19,20,21,22,23,24,25,26,27,28,29,30,31,32,33,34,35,36,37,38,39,40]. The majority of them occurred in malignant tumors, confirming the importance of histological malignancy on local prognosis [13,14,15].

Once the resection is complete, the continuity of the extensor apparatus must be reconstructed to restore its functionality and enable adequate postoperative performance to the treated knee. Due to the rarity of patellar malignant tumors, most of the evidence on surgical techniques and clinical outcomes relies on case reports [17,18,19,20,21,22,23,24,25,26,27,28,29,30,31,32,33,34,35,36,37,38,39,40]. Although the absence of large-scale case series limits the comprehension of each approach’s advantages and disadvantages compared to the others, the literature offers several different reconstructive approaches after total patellectomy. 

Some authors did not replace the patella, performing nothing but a direct suture on the remaining ends of the extensor apparatus [18,19,22,23,32,35,39,40]. This approach can be considered exclusively when the resection is confined to the bone, whereas the quadriceps tendon, the patellar ligament, and the anterior tendon fibers of the rectus femoris are completely preserved. Easy and quick to perform, simple sutures can restore the mechanical strength of the extensor apparatus enough to allow active mobilization and achieve a total or subtotal range of motion in extension. However, the absence of the patella’s mechanical properties, without a proper replacement, is insufficient to return patients to their actual pre-surgery strength [23,32,40]. Direct sutures should therefore be considered for low-demanding patients or cases with a short life expectancy that would benefit from short and less complex surgical procedures, on the condition that the patella was the only segment of the extensor apparatus to be sacrificed during the surgical procedure [18,19,35]. 

In massive lesions, pathological fractures, and most primary malignant tumors, patellectomy alone might not be sufficient to achieve wide resection margins. Since the complete local eradication of the disease is crucial for a patient’s survival, intralesional approaches cannot be accepted in the case of malignancies despite their negative mechanical and anatomical implications [42]. Therefore, surgeons sacrifice a larger share of the patient’s extensor apparatus, including the soft tissues. 

Some authors reported on using local tendon transpositions and transfers, sacrificing nearby soft tissues to reestablish the anatomical and functional continuity of the extensor chain [20,24,28,29]. Different case reports suggest that muscle and tendon plasty can lead to good functional results [24,28,29]. These approaches could represent an alternative to sutures alone in cases of short tissue gaps that cannot be directly fixed without local tissue addition. Flaps and transfers reported in the literature have shown good functional outcomes, with a complete or subcomplete range of motion, despite an inevitable downgrade in muscular strength for both the extensor apparatus and the donor muscle compared to the contralateral healthy knee [24,28,29]. Muscle and tendon plasty and flaps rely on active perfusion, which might help decrease the risk of dehiscence or infections and ease healing processes. In selected cases, using myocutaneous flaps could also be considered to provide fiber tissues to replace the patella and skin coverage after massive resections with superficial involvement [27,28]. However, these approaches come at the price of longer surgical times and the sacrifice of a donor site that might face functional impairment and can be exposed to local complications. 

Non-autologous augmentations can fulfill the massive tissue gaps resulting from massive resections for malignant patellar tumors. Massive bone and tendon allografts, including the entire extensor apparatus or its components, are among the most extensively described reconstructive approaches in the literature [15,25,30,37,38]. Once stabilized at the distal and proximal ends of what remains of the extensor chain, massive allografts that replace the patella allow for adequate transmission of the quadriceps’ strength and provide the best performance among all available techniques [15,38]. For this reason, massive allografts may represent the treatment of choice for massive resections in high-demanding patients, especially those with medium or long life expectancy. As time passes, the surfaces of the allograft are expected to be colonized by the patients’ cells and function as a scaffold for tissue integration. However, the inner parts of the extensor apparatus, particularly the inner patella and the bone—soft tissue interfaces, are rarely reached by the patient’s cells, which eventually leads to mechanical complications, including post-traumatic fractures and tears [15,45].

Furthermore, bone and tendon allografts require tissue and bone banks that are not always available worldwide. Some articles also reported cases where only soft tissue allografts, mainly fascia lata layers, were used in conjunction with autologous bone, cement, or synthetic surgical meshes to mimic the function of the native patella [34,37]. These alternatives can be used when massive bone–tendon allografts are unavailable or when surgeons aim to reduce the patellar hindrance to allow skin closure or flexion in selected cases. 

No evidence of prosthetic replacements for the patella could be found, although one paper reported on the use of synthetic tendons and meshes in association with other local augments [34,36].

A summary of the advantages, disadvantages, and proposed indications for the most used and described reconstructive approaches after patellectomy in orthopedic oncology is reported in Table 2, with a proposed classification system.

We acknowledge that our study has some limitations. The rarity of patellar tumors limited the number of case series and the size of their cohorts, thereby limiting the reliability and significance of available data. Another limitation is that, due to the nature of our review, only reconstructive options already used and described in the orthopedic oncology setting have been considered. Although the approaches included in our study are differentiated and heterogeneous, some reconstructive techniques already used in other orthopedic fields may also apply to orthopedic oncology. In particular, a variety of muscular flaps, including the reversed quadriceps tendon flap, have been described in the literature to address large gaps in infections and non-oncological settings [46,47,48]. Alongside the techniques reported in our review, these additional methods could also be considered when treating extensor apparatus defects caused by malignant bone tumors.

Despite these limitations, our review provides an unprecedented overview of surgical treatments described in the literature to reconstruct the extensor apparatus after oncologic resections. Furthermore, it categorizes reconstructive approaches, highlighting their advantages and disadvantages while also providing indications for each surgical procedure.

This review summarizes the modern literature and aims to guide orthopedic surgeons in performing the surgical approach that best suits the individual patient’s needs. In the future, the proposed guidelines may enable a more standardized approach, opening the door to a larger case series and multicenter studies on relatively homogeneous cohorts, which could increase the statistical significance of the data available today.

## 5. Conclusions

To this date, the literature lacks large-scale studies on the effectiveness of the proposed reconstructive approaches in terms of complication rates and functional outcomes. Multicenter studies would be advisable to overcome the low incidence of patellar tumors, and cohort studies could allow a comparison between the outcomes of different approaches in similar conditions. 

However, several different reconstructive approaches have already been described in the literature, and our review first summarizes their outcomes. Surgeons should be aware of each technique’s advantages and disadvantages and choose the best reconstruction for each individual, depending on the tumor type, local extension with potential soft tissue involvement, the patient’s prognosis, and functional demands. 

## Figures and Tables

**Figure 1 jcm-14-04818-f001:**
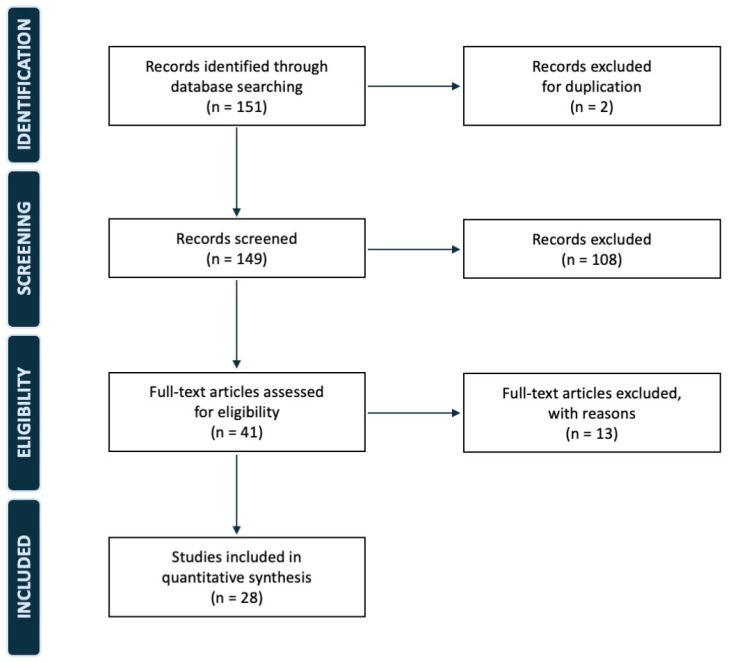
PRISMA flow diagram of our study.

**Figure 2 jcm-14-04818-f002:**
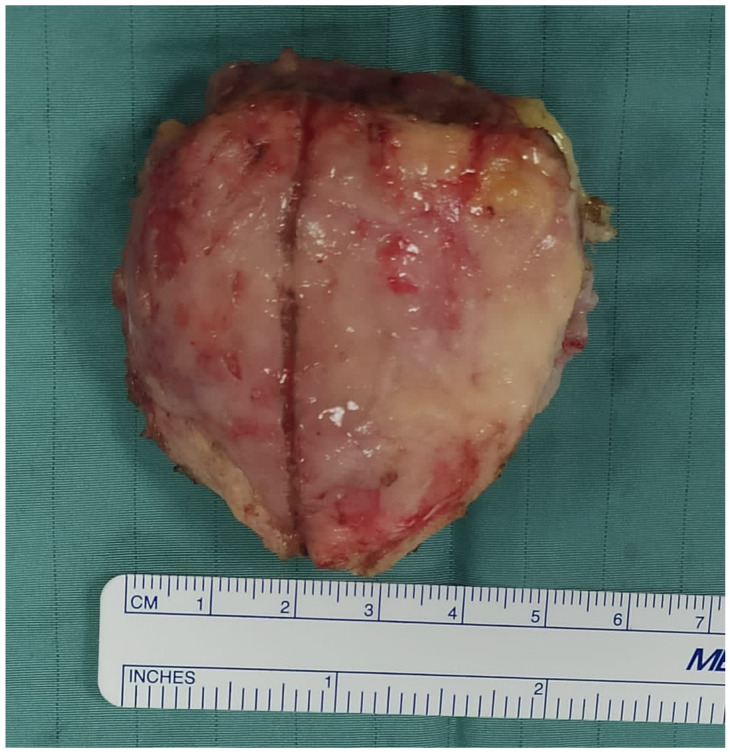
Intra-operative image of a patella enlarged and deformed by a malignant tumor after total patellectomy.

**Figure 3 jcm-14-04818-f003:**
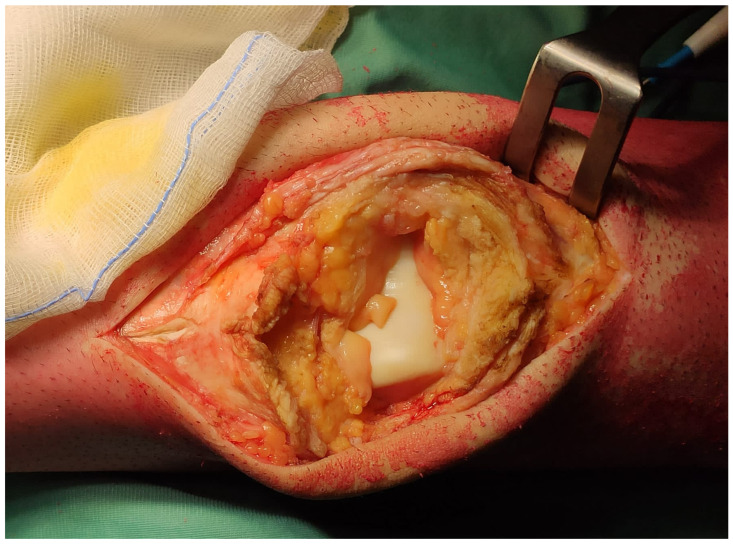
A gap in the extensor apparatus after total patellectomy associated with soft tissue sacrifice. The two ends of the damaged extensor apparatus could not be sutured directly, and the anterior articular surface of the distal femur is exposed.

**Figure 4 jcm-14-04818-f004:**
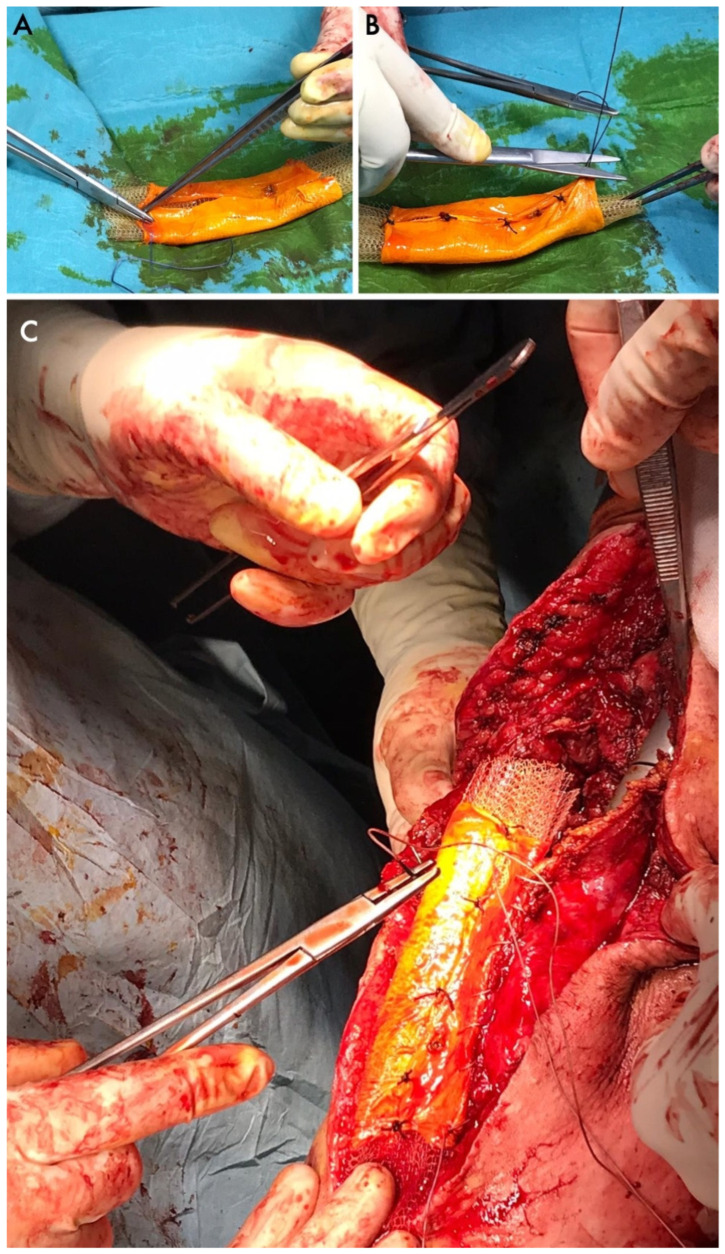
A mesh and fascia lata allograft folded (**A**) and sutured (**B**) to replace the original extensor apparatus according to the Andreani technique (**C**).

**Figure 5 jcm-14-04818-f005:**
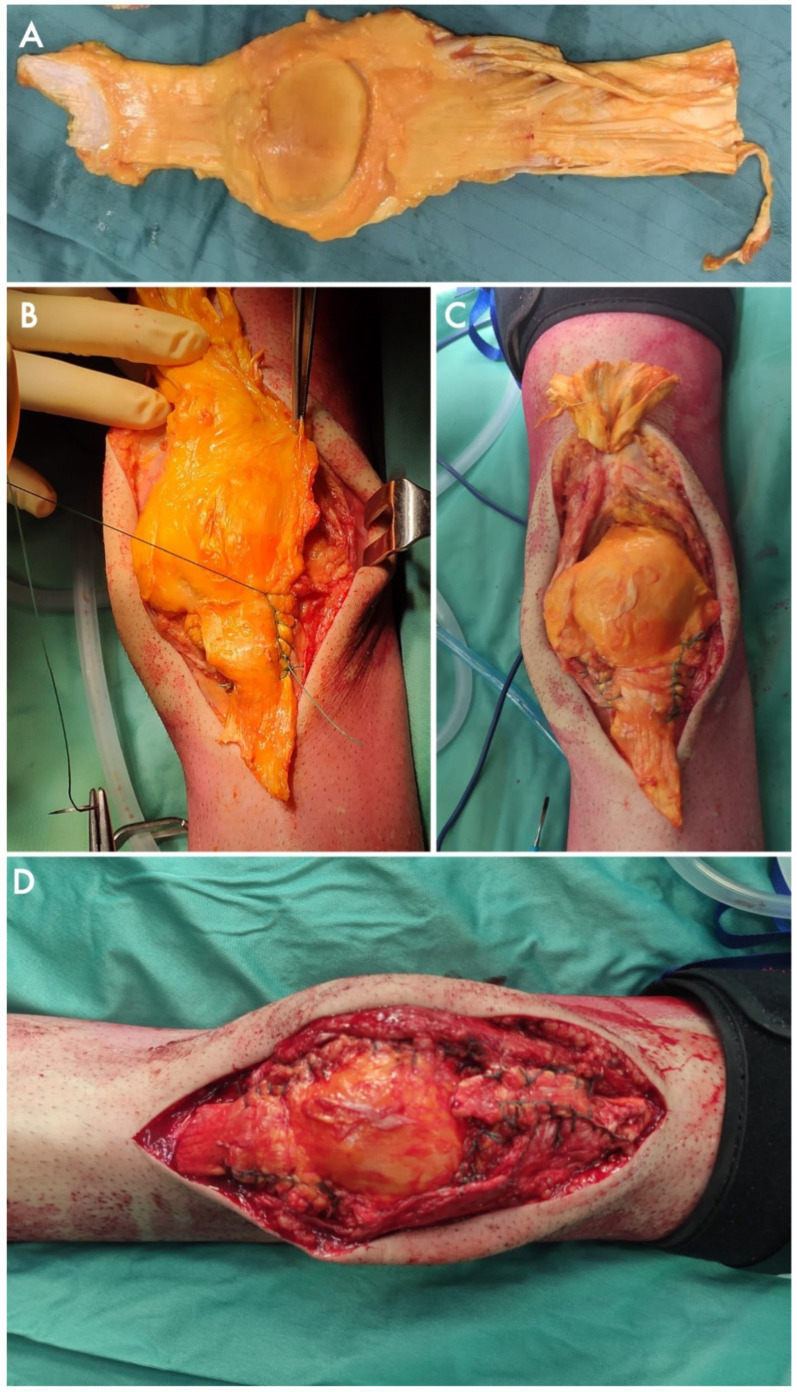
(**A**) A massive tendon-bone allograft including a quadriceps tendon (on the right), a patella (in the middle), and a patellar tendon (on the right). The graft is sutured to the patellar retinacles and the proximal tibia (**B**), and later to its proximal end, also passing through what remained of the native quadriceps tendon (**C**). The result is pictured on the bottom figure (**D**).

**Table 1 jcm-14-04818-t001:** A summary of all the articles included in our review.

Article (Year)	N	Age	Diagnosis	D	S. T. I.	Fr.	Reconstruction	LR	C	C Type	Functional Comments	MSTS	ROM	FU
Klenerman (1965) [17]	1	48	MTS Breast Carc.	-	No	No	NA	No	No	-	Walked with sticks before aggravation	NA	NA	3
Mercuri et al. (1991) [13]	7	33.7	1 ABC 5 GCT bone 1 Plasmocyt.	-	NA	7 No	NA	1 Yes 6 No	NA	-	-	NA	NA	53.7
Jaeger et al. (1992) [18]	1	65	MTS Melanoma	-	No	No	Type A	NA	Yes	Died 2 weeks after surgery	-	NA	NA	0.4
Okada et al. (1994) [19]	1	54	OS	5.0	No	Yes	Type A	No	NA	-	-	NA	NA	24
Connel et al. (1997) [20]	1	44	GCT bone	-	No	No	Type B	No	No	-	-	NA	NA	10
Aktas et al. (1998) [21]	1	55	MTS Lung Carc.	4.0	No	No	NA	No	No	-	Returned to daily activities	NA	NA	8
MacDonald et al. (2001) [22]	1	36	GCT bone	5.0	No	No	None	No	No	-	-	NA	NA	15
Bhagat et al. (2008) [14]	2	63.5	2 GCT bone		No	1 Yes 1 No	NA	No	1 Yes 1 No	1 Post-op Stiffness	1 Post-op Stiffness 1 Ok	NA	1 Limited range 1 Full range	60
Gudi et al. (2008) [23]	1	24	CB	-	No	Yes	None	No	No	-	-	NA	Full range	24
Osanai et al. (2008) [24]	1	42	MFH	6.5	Yes	No	Type B	No	No	-	-	28	5/85	12
Cho et al. (2009) [25]	1	53	OS		No	No	Type C	No	No	-	-	NA	10/140	26
Burk et al. (2010) [26]	1	13	MTS Melanoma		No	No	NA	No	No	-	-	30	Full range	42
Muramatsu et al. (2010) [27]	1	69	MFS	3.5	Yes	No	Type B	No	No	-	Extensor Apparatus MRC 4/5	NA	10/110	18
Cetinkaya et al. (2016) [28]	1	32	ABC	8.0	No	No	Type B	No	No	-	-	NA	5/105	22
Muller et al. (2018) [15]	6	-	2 MFS 2 PS 2 SS		Yes	No	Type B Type C	2 Yes 4 No	3 Yes 2 No	1 Tendon graft rupture 1 Flap neurosis 1 Bone graft fracture	-	24.7	10/82.5	80.3
Gomes et al. (2019) [29]	1	23	ABC	12.3	No	No	Type B	No	No	-	-	30	0/110	2
Gomez Palomo et al. (2019) [30]	1	24	MFS	-	No	No	Type C	No	No	-	-	NA	0/120	60
Srikant et al. (2019) [31]	1	23	FD	13	No	No	NA	No	No	-	-	NA	NA	NA
Kumar Arora et al. (2020) [32]	1	16	GCT bone	-	No	No	Type A	No	No		-	30/30	Full range	22
Cao et al. (2021) [33]	1	50	Lymphoma		No	Yes	NA	No	No		-	NA	0/135	12
Merchan et al. (2021) [16]	8	41.9	3 GCT bone 1 CB 1 CS 1 SS 1 MTS Lung Carc. 1 Lymphoma	-	NA	1 Yes 7 No	NA	8 No	NA	-	-	NA	NA	NA
Andreani et al. (2022) [34]	1	67	GCT bone	-	No	No	Type C	No	No		Extensor Apparatus MRC 5/5	30	0/120	14
Furuta et al. (2023) [35]	1	65	MTS Gastric Carc.	-	No	No	None	No	No		-	26	NA	10
Yokoyama et al. (2023) [36]	1	16	OS	-	Yes	No	Type C	No	No		Walks with orthosis, but no crutches	NA	NA	12
Chhajed et al. (2024) [37]	1	33	GCT bone	-	Yes	No	Type C	No	No		Gradual return to complete extension	30	Full range	10
Cosseddu et al. (2024) [38]	1	39	MTS Lung Carc.	-	No	No	Type C	No	No	-	Extensor Apparatus MRC 5/5	30	0/120	12
Mukherjee et al. (2024) [39]	1	20	GCT bone + ABC	-	No	No	Type A	No	No	-	Good range of motion, pain-free, no supports	NA	NA	12
Sakuda et al. (2024) [40]	1	74	Myopericyt.	2.0	No	No	Type A	No	No	-	Extensor Apparatus MRC 3/5	NA	10/110	36

N = Number. D = Diameter. S.T.I.= Soft tissue involvement. FR = Pre-operative Fracture. LR = Local recurrence. C = Complications. ROM = Range of motion. FU = Follow up. Carc = Carcinoma. MTS = Metastasis. OS = Osteosarcoma. ABC = Aneurysmal bone cyst. GCT = Giant cell tumor of bone. CS = Chondrosarcoma. CB = Chondroblastoma. FD = Fibrous dysplasia. SS = Synovial sarcoma. Myopericyt = Myopericytoma. MFS = MyxoFibroSarcoma. PS = Pleomorphic sarcoma. Plasmoccyt. = Plasmocytoma. MFH = Malignant fibrous histiocytoma. Type A = Direct suture. Type B = Tendon plasty, transfers, and/or muscle flaps. Type C = Allografts/synthetic augments.

**Table 2 jcm-14-04818-t002:** A schematic resume of the most commonly used reconstructive approaches to replace patella after total patellectomy in orthopedic oncology. The advantages, disadvantages, and indications of each approach are summarized.

Reconstruction	Advantages	Disadvantages	Indications
**TYPE A** Direct suture	Less surgical time No augments needed	Soft tissues of the extensor apparatus must be preserved Limited extensor apparatus’ strength and clinical performances	Preserved soft tissues of the extensor apparatus and - Short life expectancy - Low demanding patients
**TYPE B** Tendon plasty/Transfers/ Muscle flaps	Autologous tissue and active blood supply May include skin coverage	Longer surgical times Involvement of a donor site	Gaps in the extensor apparatus that could not be sutured, and - Long life expectancy - Low demanding patients
**TYPE C** Allografts/ Synthetic augments	Mechanical strength and functional performances Can fulfill large gaps without a donor site	Require bone and tissue banks Risk of mechanical complications and failures	- High demanding patients - Mid or long life expectancy

## Data Availability

Data availability statement: The data that support the findings of this study are available from the corresponding author upon reasonable request.

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
