# Peer review of "Reconstruction of the Extensor Apparatus After Total Patellectomy in Orthopedic Oncology: A Systematic Literature Review"

_jcm, 2025, doi:10.3390/jcm14144818_

Round 1
Reviewer 1 Report
Comments and Suggestions for Authors
This is an informative and clinically useful review on a rare but challenging situation. I personally find the proposed classification and decision-making strategy summarized in Table 2 to be very helpful. It gives a structured and practical overview for reconstructive choices after total patellectomy, which is particularly valuable given the limited literature.
- The databases and example keywords are mentioned; however, the full search strings, use of MeSH terms, final search dates per database, and treatment of grey literature are not described. I recommend including this information in an Appendix.
- Table 1 is hard to read. Please consider increasing the font size or improving the layout for better readability.
- In line 180, “MCR score” might be a typo and should probably be corrected to “MRC score”.
- References 25 and 33 may refer to the same article.
- The manuscript does not currently include a discussion of its limitations. Please add a brief section that outlines the study's main limitations or issues.
- The review lacks any mention of future research directions. Please include a short paragraph suggesting areas where further investigation would be valuable.
Author Response
Dear Reviewers and Editorial Board,
Thank you for your help and your suggestions to increase the quality of our paper.
What follows are replies to the questions and suggestions made by our reviewer. All the corrections and the new sentences are written in red in our revised manuscript.
1.
As suggested, we added an appendix (lines 73; 360-365) to report in detail the search strings used in our research and the date on which we performed our string research. No grey literature was included, as the study was carried out as already reported in lines 69-96.
2.
We agree that the previous version of Table 1 was difficult to read, and we have increased the text size from 5.5 to 7.5.
3.
Yes, the typo in line 180 was a mistake that has now been corrected.
4.
Although the reference for article 25 (Cho Y et al., 2009, doi: 10.3928/01477447-20090818-27) was correct throughout the manuscript, we made an error by replacing it in the reference list with another article (Cao et al., 2021). Thank you for the advice; the error has been corrected.
5 and 6.
New paragraphs regarding our article’s limitations and future research directions have been included in the final part of our discussion (lines 314-333).
Reviewer 2 Report
Comments and Suggestions for Authors
This literature review studied the reconstruction methods of the extensor system after patellar resection. The author's topic was very tricky, and it is admirable that such a complete system can be summarized in this small disease. This issue is a clinically appropriate direction for exploration. The author needs to provide more practical operation illustrations to confirm the author's previous work and the inductive ideas based on it. This is more conducive to us appreciating this difficult work. It is recommended to revise
Comments on the Quality of English Language
This literature review studied the reconstruction methods of the extensor system after patellar resection. The author's topic was very tricky, and it is admirable that such a complete system can be summarized in this small disease. This issue is a clinically appropriate direction for exploration. I support the publication of this article, but before publication, the author needs to provide more practical operation illustrations to confirm the author's previous work and the inductive ideas based on it. This is more conducive to us appreciating this difficult work. It is recommended to revise and publish it
Author Response
Dear Reviewers and Editorial Board,
Thank you for your help and your suggestions to increase the quality of our paper.
What follows are replies to the questions and suggestions made by our reviewer. All the corrections and the new sentences are written in red in our revised manuscript.
We agree that more figures could increase readers’ comprehension of the topic. For this reason, in line 153, we added a composite picture as a new Figure 4 to illustrate one of the techniques. We also included a new Figure 5, which not only pictures a massive extensor apparatus allograft but also its intraoperative use, replacing a wide defect of the extensor apparatus.
We also tried to fix some typos in our manuscript to increase the quality of our English.
Reviewer 3 Report
Comments and Suggestions for Authors
The framework of manuscript is intact and English writing is fluent. Articles reporting similar conditions are few (most of all are case reports) and treatment techniques are markedly varied. This article may therefore provide a valuable choice for surgeons under similar conditions.
Some doubts still require clarification:
- L-60: Is [little] or [few] more suitable in use for counting?
- Figure 1: IDENTIFICATION, n=2 for exclusion. Typing is erroneous for [excluded].
- Words used in the text should be consistent. L-72, Figure 1; L-95, Fig. 2; L-130, Fig. 3; L-155, Fig. 4.
- L-111: Full spelling of MSTS, ISOLS?
- Dates of data collection are contradicted. L-14, 1950~2024; L-78, 1964~2024; L-113, 1963~2024?
- L-217: Patellofemoral or femoropatellar joint?
- After L-242~254, please additionally comment the possibility of using reversed quadriceps tendon flap for bridging the large gap, which can avoid suturing tightness of the extensor apparatus. This technique is similar to the reversed Achilles tendon flap to treat Achilles tendon rupture with retraction (e.g., Lindholm technique, or ---).
- In Conclusions of Abstract and Conclusions of the text, please consider to suggest the reversed quadriceps tendon flap technique in order to help readers to improve surgical outcomes after total patellectomy.
- References should be written in a consistent form, e.g. capital (Refs. 1,8,15,16,25,29,30,33,34,36,37,38,39,41,43,44) or lowercase (Refs. others)?
Author Response
Dear Reviewers and Editorial Board,
Thank you for your help and your suggestions to increase the quality of our paper.
What follows are replies to the questions and suggestions made by our reviewer. All the corrections and the new sentences are written in red in our revised manuscript.
As suggested, we introduced the word few in line 60.
Figure 1 has been corrected, and the typo has been fixed.
As required, we replaced 'Figure' with 'Fig.' in line 72, providing consistency throughout the text.
We explained the meaning of the acronyms MSTS and ISOLS (lines 111-112).
The data discrepancy has been corrected. The correct timeline was 1950-2024.
We confirm the use of the term 'patellofemoral', as it is the most commonly used in English literature.
7 and 8.
We added the potential use of muscle flaps that have not been previously utilized in orthopedic oncology, including the reversed quadriceps tendon flap. We did not include it within the central part of the discussion, nor in our abstract or our conclusions, as it falls beyond the actual results of our review). However, we mentioned these techniques in the final part of our new discussion (Lines 318-324).
All references previously written with capital letters are now written in lowercase.